# Prevalence of Different *Salmonella enterica* Subspecies and Serotypes in Wild Carnivores in Emilia-Romagna Region, Italy

**DOI:** 10.3390/ani12233368

**Published:** 2022-11-30

**Authors:** Lorenzo Gambi, Valentina Ravaioli, Rachele Rossini, Vito Tranquillo, Andrea Boscarino, Sara Mattei, Mario D’incau, Giovanni Tosi, Laura Fiorentini, Alessandra Di Donato

**Affiliations:** 1Istituto Zooprofilattico Sperimentale Della Lombardia e dell’Emilia Romagna (IZSLER), 25124 Brescia, Italy; 2Istituto Zooprofilattico Sperimentale Della Lombardia e dell’Emilia Romagna (IZSLER), 47122 Forlì, Italy; 3Public Health Unit of Cesena, U.O.D. Sanità Animale, 47521 Cesena, Italy

**Keywords:** *Salmonella* spp., wild carnivores, badger, red fox, wolf

## Abstract

**Simple Summary:**

*Salmonellae* are enteric bacteria capable of infecting humans and both domestic and wild animals. Even if salmonellosis in humans is generally transmitted through food, the role of wildlife in the ecology of this bacteria is of increasing interest because of their potential role as reservoirs. In Italy, and in particular in the Apennines area, the interactions between wildlife and humans, pets and livestock are rising due to a growing wildlife population and this poses the problem of accidental *Salmonella* infections. The aim of this study was to investigate the presence of *Salmonella* subspecies and serotypes in wild carnivores (namely red fox, European badger and wolf) from the Emilia-Romagna Region between 2016 and 2022. Samples of the large intestine were cultured, and both serogroup and serotype identification were performed. A total of 67 strains were isolated, belonging to *S. enterica* subsp. *enterica*, *S. salamae*, *S. diarizonae* and *S. houtenae*. The most frequently isolated serotypes were S. veneziana and S. typhimurium. These findings highlight a prevalence of *Salmonella* spp. in line with other studies, showing once again the value of monitoring different possible sources of *Salmonella* infection.

**Abstract:**

*Salmonella* is a pathogen of considerable health concern, given its zoonotic potential, and, in Italy, is the most frequently reported causative agent for foodborne outbreaks. Wild animals and in particular wild carnivores may be carriers of different *Salmonella enterica* subspecies and serotypes. Given their potential role as reservoirs, surveillance activities are necessary. This study aims to investigate the presence of different *Salmonella* subspecies and serotypes in wild carnivores in the Emilia-Romagna Region. A total of 718 fox (*Vulpes vulpes*), 182 badger (*Meles meles*) and 27 wolf (*Canis lupus*) carcasses, submitted between 2016–2022, were included for the present work. Gender and age data were collected along with geographical coordinates of carcass’ discovery site. Contents of the large intestine were sampled and cultured according to ISO 6579-1 and both serogroup and serotype identification were performed according to ISO/TR 6579-3:2014. *Salmonella* was retrieved from 42 foxes (6%), 21 badgers (12%) and 3 wolves (12%), respectively. Isolated *Salmonella enterica* strains belonged to 4 different subspecies and 25 different serotypes. S. veneziana and S. typhimurium were the most frequent serotypes found (11/67 and 10/67, respectively). In conclusion, zoonotic serotypes were found in all these species of wildlife, thus confirming their potential role in the ecology of *Salmonella* spp.

## 1. Introduction

*Salmonella* is a genus of Gram-negative ubiquitous bacteria belonging to the *Enterobacteriaceae* family. Two species are known, namely *S. bongori* and *S. enterica*; the latter is further divided into six subspecies, called *enterica, salamae, arizonae, diarizonae, houtenae* and *indica* [1]. To date, more than 2600 serotypes are known: some are species-specific, while others have the ability to infect a wide range of species [2]. The main route of infection is oro–fecal, and some important sources of contamination are food, water, and the environment. *Salmonella* is the second most frequently reported causative agent of foodborne outbreaks and, in most cases, humans acquire the infection through foods such as meat, dairy products and eggs, which can be contaminated during the various stages of farming or during preparation and storage of food [3]. However, cases of interhuman transmission, or by direct contact with animals can occur [4]. Several studies have highlighted the ability of *Salmonella* to infect both warm-blooded and cold-blooded domestic animals and wildlife, which can act as a reservoir for serotypes related to humans’ infections [5,6,7]. Wild animals may harbor different *Salmonella enterica* subspecies and serotypes and, in particular, wild carnivores due to their nutrition habits. Predators may interact with livestock, pets and humans, generating accidental *Salmonella* infections, thus posing a danger in different occasions. On the other hand, interactions between wildlife and human-related activities may result in infections for wild animals, threatening the numbers of various wild populations. Many studies in literature have described *Salmonella* infections in predators such as alligators and caimans [8,9] raptors [10,11] and mammalian predators such as foxes and badgers [12,13]. In Italy, and in particular in the Apennines area, interactions between wildlife and human activities are more frequent due to a high population density. As a matter of fact, in recent years the number of wolves has been increasing significantly as estimated, for example, through major monitoring conducted by the Italian National Institute for Environmental Research and Protection (*Istituto Superiore per la Protezione e la Ricerca Ambientale*, ISPRA) [14]. The present study aims to investigate the presence of different *Salmonella* subspecies and serotypes in wild carnivores in the Emilia-Romagna Region.

## 2. Materials and Methods

### 2.1. Sample Collection 

Between January 2016 and September 2022, 927 wild mammalian carnivores, namely 718 foxes (*Vulpes vulpes*), 182 badgers (*Meles meles*) and 27 wolves (*Canis lupus*), were analyzed for the presence of *Salmonella* spp. in the south-eastern part of the Emilia-Romagna Region (Italy) according to a non-probabilistic sampling method [15]. Species considered were mostly found dead (road kills, poisoning or natural death) or hunted, and subjected to national and regional health monitoring programs. 

Dead animals, delivered to the Diagnostic Laboratory of Istituto Zooprofilattico Sperimentale della Lombardia e dell’Emilia-Romagna in Forlì (FC, Italy), were submitted to a complete necropsy, where contents of the large intestine were sampled and cultured according to the International Organization for Standardization (ISO). Age (i.e., young, adult and not recorded) and gender (i.e., male, female and not recorded) were also recorded during the anatomopathological examination. Geographical coordinates were registered and used to construct a map representing the sampling sites of the carcasses examined and included in the study.

### 2.2. Salmonella spp. Isolation 

The isolation of *Salmonella* spp. was performed according to ISO 6579:2002/Amd 1:2007 method (ISO 2007) for *Salmonella* spp. [16].

Briefly, 25 g of intestinal contents were transferred to sterile sampling bags with 225 mL of buffered peptone water and incubated at 37 °C for 24 h, as a pre-enrichment phase. Thereafter, 0.1 mL were inoculated on a Modified Semisolid Rappaport Vassiliadis (MSRV; Oxoid, Hampshire, UK) media and incubated for 48 h at 41.5 °C. *Salmonella* spp. suspected colonies were then plated on two selective solid media: Xylose Lysine Deoxycholate agar (XLD; bioMérieux, Bagno a Ripoli, Italy) and brilliant green agar (BGA; Vacutest Kima, Arzergrande, Italy) at 37 °C for 24 h. All presumptive *Salmonella* spp. isolates were confirmed using suitable biochemical tests (Microgen GNA ID System, Microgen Bioproducts Ltd., UK). 

Serological confirmation of the *Salmonella* strains, from pure culture and removing self-agglutinating strains, were performed through rapid slide agglutination test with proper anti serum for the detection of somatic (poly O, Rabbit antiserum, SSI DIAGNOSTICA, Denmark) and flagellar (poly H, Rabbit antiserum, SSI DIAGNOSTICA, Denmark) antigens (Figure 1). 

### 2.3. Serogroup and Serotype Identification

*Salmonella* spp. serogroup and serotype identification was performed according to ISO/TR 6579-3:2014 [17]. 

The complete serological characterization of *Salmonella* was performed by slide agglutination for the determination of somatic antigens, while, for the determination of flagellar antigens, the method of tube agglutination according to the technique of Spicer (1956), modified by Edwards (1962) and Morris et al. (1972) was followed (Figure 1) [18,19,20]. 

The results of the determinations of the antigens were then used for the final serological characterization according to the scheme of Kauffmann–White–Le Minor [21]. 

### 2.4. Data Analysis

For all species, the point and interval estimates of the overall and sex/age-specific prevalence of the *Salmonella*-positive samples were obtained using a conjugate beta prior on the distribution of p (probability of being a *Salmonella*-positive sample) in a binomial experiment obtaining an a posteriori Beta distribution of the probability p [22]. From Bayes’ theorem the posterior distribution of p given the observed data is: p|x ~ Beta (x +α, n − β)
where: p = probability to be *Salmonella* spp.-positive; x = number of positive samples; n = number of tested samples; α and β are the hyperparameters of the a priori beta distribution of p, which in this case is a Jeffrey’s prior with distribution Beta (0.5, 0.5).

From the posterior distribution, a credibility interval is then constructed which collects the highest probability of density (HPD) corresponding to 95% of the probability (**p**) values estimated. 

Point and interval estimates of *Salmonella* spp. prevalence were calculated using function binom.bayes of the binom package [23] in R language [24]. All results were presented with a forest plot for each species, obtained by using the forestplot function of the forestplot package [25] in R. A map was drawn using leaflet and sf packages [26,27]. 

## 3. Results

The sampling sites, shown in Figure 2, reflect the geographical area of sampling in dependence on the spatial competency of the headquarters that conducted the study. 

Wolves and badgers were, in the most cases, road traffic accident victims, thus being submitted to passive surveillance. On the other hand, foxes were mainly hunted and thus being subjected to active surveillance. No salmonellosis-related lesions were identified during necropsies.

### 3.1. Salmonella Prevalence

A total of 66 out of 927 samples tested positive for *Salmonella* spp. The overall prevalence in the different species was found to be, respectively, 6% in foxes (95% CI: 4–8%), 12% in badgers (95% CI: 7–16%) and 12% in wolves (95% CI: 2–25%) (Figure 3). 

A total of 371 carcasses were recorded as female (285 foxes, 74 badgers and 12 wolves), 416 were recorded as male (329 foxes, 72 badgers and 15 wolves), while no registration was performed for 140 carcasses (104 foxes, 36 badgers). Moreover, age registration pointed out a total of 606 adult animals (456 foxes, 137 badgers, 13 wolves), 259 young animals (220 foxes, 25 badgers and 14 wolves) and 62 unrecorded (42 foxes and 20 badgers). 

Concerning badgers, the point estimate of prevalence ranges from a lowest of 4% (95% CI: 0–13%) in young females to 25% (95% CI: 3–50%) in young males. Isolates from wolves belonged to one adult female (19%; 95% CI: 0–44%), one adult male (21%; 95% CI: 0–50%) and one young female (25%; 95% CI: 0–56%), while all young males were found negative. Age and gender results in foxes displayed better confidence intervals, thanks to a large number of samples. Adult and young female foxes both presented a prevalence of 5% (95% CI: 2–9% and 95% CI: 2–10%, respectively), adult males 8% (95% CI: 4–11%) while young males 9% (95% CI: 4–14%) (Figure 3).

### 3.2. Salmonella Species and Serotypes

A total of 67 *Salmonella* isolates, with 25 different serotypes, were detected from 42 foxes, 21 badgers and 3 wolves, respectively. Only one animal, a badger, showed two serotypes of *Salmonella* (*S. umbilo* and *S. typhimurium*) in the same matrix.

Overall, four subspecies were detected: *S. enterica* subsp. *enterica* (61 strains, 22 serovars), *S. enterica* subsp. *salamae* (3 strains, 1 serovar), *S. enterica* subsp. *houtenae* (1 strain, 1 serovar) and *S. enterica* subsp. *diarizonae* (2 strains, 1 serovar). The most common isolated strain for *S. enterica* subsp. *enterica* was S. veneziana (n = 11), followed by S. typhimurium (n = 10) and S. newport, which was retrieved in six cases. Table 1 gives an overview of the other serotypes identified.

While badgers and wolves showed only *S. enterica* subsp. *enterica*, in foxes’ samples all four subspecies and 23 different serotypes were observed.

*S. typhimurium* and its monophasic variant were found in foxes in five samples and once, respectively, while five *S. typhimurium* isolates and two *S. enteritidis* isolates were retrieved from badgers.

Concerning wolves, two animals were found positive for *S. infantis* and one for *S. stanleyville*.

## 4. Discussion

*Salmonella enterica* is a zoonotic and food-borne pathogen for hundreds of vertebrate species, including human beings; thus, interspecies transmission can occur in many different ways. For example, wildlife may become infected because of livestock, domestic animals or human wastes, or there may be transmission between wild animals [28,29]. In this framework, wild carnivores may have a crucial role in spreading *Salmonella* in the environment or getting infected due to their nutrition habits as scavengers or predators [30,31]. The current study showed the presence of different *Salmonella* subspecies and serotypes in wild mammalian carnivores from the Emilia-Romagna Region between 2016 and 2022. Three wildlife species were included, namely red fox, European badger and wolf. Other local wild mammalian carnivores, such as stone martens (*Martes foina*), weasels (*Mustela nivalis*), polecats (*Mustela putorius*) and wild cats (*Felis silvestris*) were excluded from the study due to the small number of carcasses submitted to the laboratory in the same period. The low frequency of carcasses’ submission might be related to the animals’ habits and physical features: as they are small, mostly nocturnal mammals with elusive behavior, so it may be more difficult to find carcasses related to road killing or poisoning. On the other hand, wolves, red foxes and European badgers are the biggest carnivores of the Emilia-Romagna Apennines mountains and in the last 20 years all populations have been increasing in numbers [14,32,33]. That leads to more contact with human settlements, and therefore deaths due to human-related activities. 

No lesions related to salmonellosis were identified during necropsy. This finding suggests that most infected animals might be asymptomatic carriers. A total of 67 different *Salmonella* isolates were obtained in the present work, with a prevalence of 6% (95% CI: 4–8%) and 12% (95% CI: 7–16%) in foxes and badgers, respectively. These results are in line with most of the European studies in literature, where prevalence ranges from 0% (95% CI: 0–2%) to 10% (95% CI: 0–25%) for foxes and between 7% (95%CI: 6–8%) and 28% (95% CI: 20–36%) for badgers (Appendix A). Moreover, the prevalence highlighted in Italian studies with good confidence intervals are equivalent to our findings [12,13,34,35]. For further details, refer to Appendix A. 

Regarding the isolated subspecies, 61 strains (92.4%) belonged to *S. enterica* subsp*. enterica*, while the remaining 6 strains, respectively, were *S. salamae* (3/6), *S. diarizonae* (2/6) and *S. houtenae* (1/6). Isolates from badgers and wolves belonged only to the first subspecies, while the latter three were identified in foxes. Those were the same subspecies isolated in other European studies regarding foxes, with similar low prevalence compared to *S. enterica* subsp*. enterica* [12,28,34,35,36,37]. Furthermore, Guidetti et al identified *S. diarizonae*, *S. houtenae* and *S. salamae* also in mustelids, without specifying whether they were isolated from badgers, pine marten, stone marten or polecats [35]. Lesser *Salmonella* subspecies are frequently isolated from reptiles and birds, which may become the source of infection for wildlife, domestic mammals and humans. In particular, infection in wild carnivores is probably related to predation and scavenging of reptiles and birds [11,38]. Concerning *S. enterica* subsp. *enterica* serotypes, the most frequent was *S. veneziana* with 11 isolates (16.4%). Despite being already reported from both foxes and badgers, this is the highest prevalence retrieved from literature [12,13,34]. As *S. veneziana* was recently isolated from wild boars in Italy [39,40], the high prevalence described in the present work might be related to sharing of the same environment and to scavenging of wild boars’ carcasses. The second most frequently isolated serotype was *S. typhimurium* with 10 strains. This finding is in line with other surveys where this serotype was commonly isolated [13,41,42]. It might be related to the fact that *S. typhimurium* is frequently associated with wild and domestic birds, which share the same environment with foxes and badgers and are part of their nutrition habit [43,44]. Moreover, a total of six *S. newport* strains were identified in the present study from foxes and badgers. This serotype was already isolated in the last 20 years in Europe from the same wild carnivores, suggesting a permanent presence of *S. newport* in the environment [37,45,46]. Interestingly, *S. agama* was retrieved only in two foxes and two badgers. This specific serotype has been considered typical of badgers in the British and Irish islands for the last 45 years [41,46,47,48]. The low prevalence of the present work might suggest that it is not widespread in continental Europe. At last, no *S. dublin* could be retrieved, unlike what Glawischnig at al described in Austria [28]. This feature could be related to the absence or low prevalence in the Apennines of this specific well-known serotype from livestock and wild ruminants. Furthermore, *S. typhimurium* and its monophasic variant, *S. enteritidis*, *S. derby* and *S. infantis* were all retrieved during the present survey. These serotypes are the top-five isolated *Salmonella* serotypes involved in human cases of salmonellosis in EU in 2020. Other relevant zoonotic *Salmonella* serotypes, such as *S. newport*, *S. coeln*, *S. muenchen*, *S. agona* and *S. kottbus* were found in the present work, where they resulted in about 47% of the total isolates. [3]. This finding highlights once more the importance of monitoring different possible sources of *Salmonella* infection for humans and domestic animals. On the other hand, a potential contamination of the environment by human wastes might be related to the presence of zoonotic *Salmonella* serotypes in wildlife [49]. No relevant differences were highlighted concerning prevalence in different ages and genders for badgers and wolves (Figure 3). These results are due to low number of subjects for each category, generating large confidence intervals. Concerning foxes, despite having more data, few differences were pointed out between different gender and age prevalence. The only relevant difference was a prevalence of 8% (95%CI: 4–11%) in adult male foxes and 9% in young males (95%CI: 4–14%), slightly higher than the one of adult (5%; 95%CI: 2–9%) and young (5%; 95%CI: 2–10%) female foxes. Male foxes, and in particular adult ones, usually have a wider dietary diversity than females, which might mean more animal species hunted or scavenged, thus a higher probability of infections with various *Salmonella* serotypes [50]. 

As can be seen from Figure 2, most of the carcasses were retrieved from a 20 km-wide strip, bordered on its northeast side by a highway, while on the southwest by the Apennines. In this territory, the human population density is high, and lessens in the Apennines. Human settlements and high population density close to a mountain forest environment result in more human–wildlife interactions, such as road killing or accidental poisoning. This leads also to a higher probability of identification of carcasses. Moreover, the rising population of many wildlife species in Italy results in more interactions, as wildlife approaches human settlements in order to find new sources of food. For example, it is interesting to see from Figure 2 how many carcasses, including two wolves, were found beyond the highway in the Padan plain, a completely different environment from the mountain forest.

Wolves are the mammalian apex predators in the Apennines, along with bears. The latter are retrieved only in the Abruzzo region, while wolves cover the whole mountain range. Many studies were carried out regarding the possible role of wolves in wildlife diseases transmission [51]. Most of the literature is related to case studies of single animals or natural park populations, and there are few studies regarding a whole territory population, where various packs dwell [52,53,54]. Furthermore, no study could be retrieved from the literature regarding *Salmonella* presence or prevalence in wolves. In the present study, three isolates were obtained from the 27 specimens, two of which were *S. infantis* and one *S. stanleyville*. As mentioned before, *S. infantis* has a well-known zoonotic potential, as it is listed as the fourth most commonly reported serovar among all confirmed salmonellosis cases from humans [3]. On the other hand, *S. stanleyville* is rarely related to human outbreaks [55] and it was mostly retrieved from pigs and wild boars [39,40]. These results, despite the low number of samples, suggest that wolves, as an apex predator in any environment, may be exposed to many *Salmonella* serotypes.

## 5. Conclusions

The present study highlighted a prevalence of *Salmonella* spp. in line with other European studies, with interesting results regarding serotypes. Moreover, data about *Salmonella* presence in wolves are a useful basis in order to identify their potential role in *Salmonella* infections in wildlife, or for a better understanding of the effects of salmonellosis on wolves’ populations. Further analysis should be carried out to assess the actual ecology of *Salmonella* spp. in predators; for instance, by relating strains isolated from wild carnivores with those from other wildlife species, livestock, pets and humans. Technologies such as whole genome sequencing will help in reaching such a goal. Moreover, analysis of the antimicrobial resistance of the isolates could help to improve the knowledge of multi-resistant circulating strains, thus understanding the risks for wildlife, livestock and humans. 

## Figures and Tables

**Figure 1 animals-12-03368-f001:**
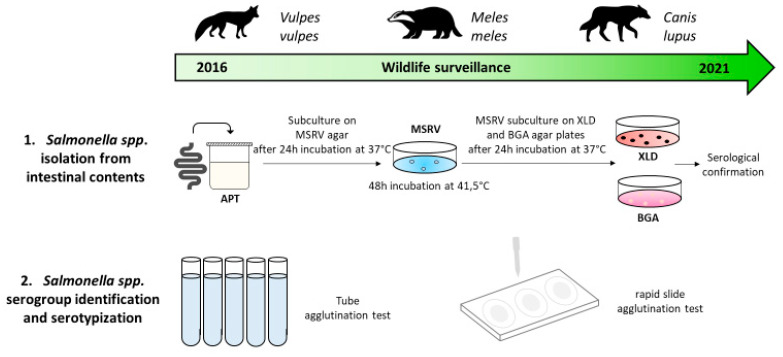
Graphic representation of *Salmonella* spp. isolation and typization procedure.

**Figure 2 animals-12-03368-f002:**
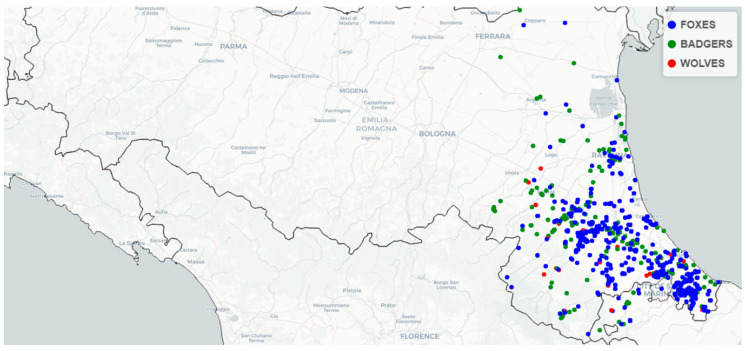
Spatial distribution of collected badger, fox and wolf carcasses (January 2016–September 2022).

**Figure 3 animals-12-03368-f003:**
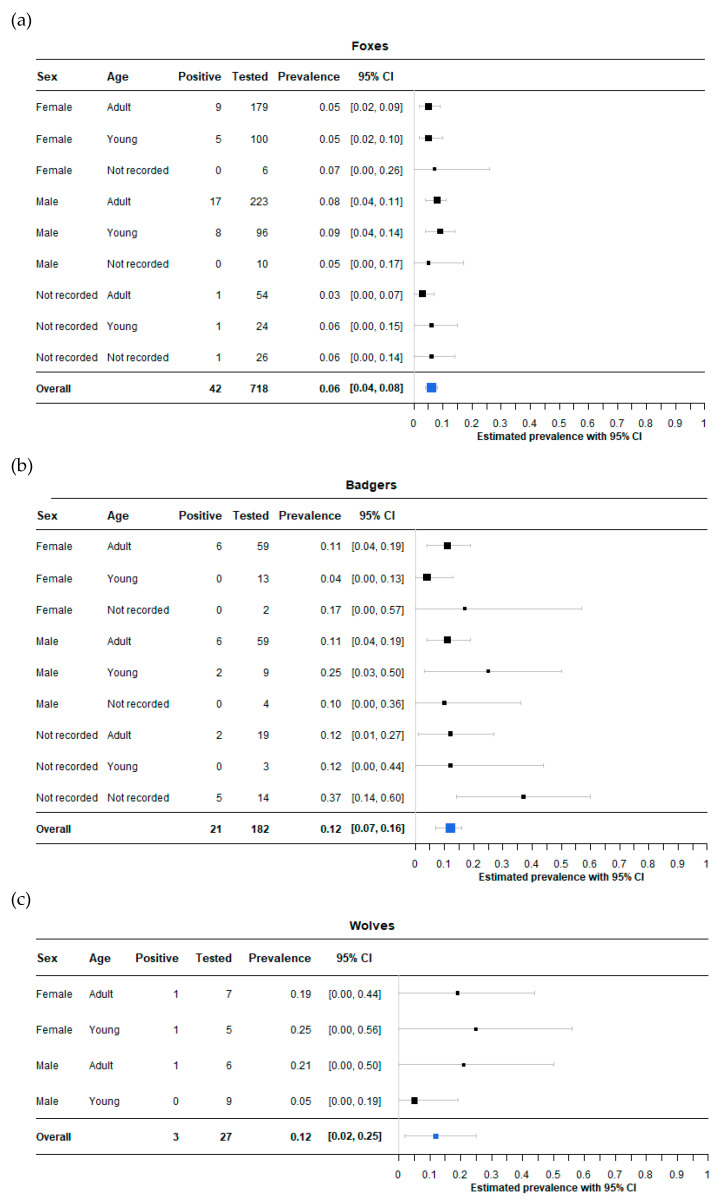
Number of the tested and positive foxes (**a**), badgers (**b**) and wolves (**c**) divided by age and gender and respective Bayesian posterior estimated prevalence with 95% credibility interval.

**Table 1 animals-12-03368-t001:** *Salmonella* subspecies and serotypes isolated from foxes, badgers and wolves.

*S. enterica serovars*	Foxes	Badgers	Wolves	Total
*S. enterica* subsp. *enterica*	36	22	3	61 (92.4%)
Veneziana	7	4	0	11 (16.4%)
Typhimurium	5	5	0	10 (14.9%)
Newport	2	4	0	6 (8.96%)
Agama	2	2	0	4 (5.97%)
Coeln	2	2	0	4 (5.97%)
Infantis	2	0	2	4 (5.97%)
Zaiman	1	2	0	3 (4.48%)
Enteritidis	0	2	0	2 (2.99%)
Farmingdale	2	0	0	2 (2.99%)
Muenchen	2	0	0	2 (2.99%)
Stanleyville	1	0	1	2 (2.99%)
Agona	1	0	0	1 (1.49%)
Anatum	1	0	0	1 (1.49%)
Bredeney	1	0	0	1 (1.49%)
Derby	1	0	0	1 (1.49%)
Give	1	0	0	1 (1.49%)
Kottbus	1	0	0	1 (1.49%)
Livingstone	1	0	0	1 (1.49%)
Mikawasima	1	0	0	1 (1.49%)
Rissen	1	0	0	1 (1.49%)
Typhimurium monophasic variant	1	0	0	1 (1.49%)
Umbilo	0	1	0	1 (1.49%)
*S. enterica* subsp. *salamae* (1 serovar)	3	0	0	3 (4.48%)
*S. enterica* subsp. *diarizonae* (1 serovar)	2	0	0	2 (2.99%)
*S. enterica* subsp. H*outenae* (1 serovar)	1	0	0	1 (1.49%)
Total	42	22	3	67

## Data Availability

The data presented in this study are available on request from the corresponding author.

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
