# Peer review of "Prevalence of Different Salmonella enterica Subspecies and Serotypes in Wild Carnivores in Emilia-Romagna Region, Italy"

_animals, 2022, doi:10.3390/ani12233368_

Round 1
Reviewer 1 Report
1.- In materials and methods it is not described how the containment of the animals was carried out and if there was a permit for it, so it is understood that part of the animals were captured and others hunted, some carcasses found and the necropsy was carried out. I don't know if some type of permit is required for its collection or ethical permission
2.- Some letters in figure 1 are not very clear
3.- Figure number 3 cannot be seen, has poor quality, and is not legible or difficult to read
4.-The reference 55 cited in the text is not found if in the bibliography
Reviewer 2 Report
Comments and Suggestions for Authors:
Plenty of literature on the prevalence of Salmonella spp. in humans and livestock is available. On the contrary little is known, and scarce data are available about the natural occurrence and spread of this pathogen in wild carnivores. In my opinion, this is an interesting manuscript that characterizes the presence of Salmonella subspecies and serotypes in wild carnivores from Europe. This manuscript provides valuable information for understanding the role of these species in Salmonella spp. ecology.
However, some modifications to the manuscript are necessary.
- It is mentioned that the study was carried out in the Emilia-Romagna Region, which is a vast territory. Figure 2 shows that only one area in the southeast of the Emilia-Romagna Region was analyzed. It is suggested that this be mentioned in the material and methods section or, in any case, in the Title to avoid confusion.
- It should be revised throughout the manuscript that Salmonella is written in italics.
- Line 67, remove "carnivores," as it confuses the reader.
- Line 72, please indicate what ISPRA stands for.
- Line 111, the reference of "ISO/TR 6579-3:2014" is missing.
- Line 129, numero or number?
- Line 137, Forrest plot or forest plot?
- Line 167, the image's resolution in Figure 3 needs to be improved, as the scale of the Bayesian representation loses its definition.
- Line 326, check the format of reference 1; please homologate it.
Reviewer 3 Report
I thoroughly enjoyed reading the manuscript investigating Salmonella spp., in wild carnivores in Italy. It is relatively rare to see the investigation of Salmonella spp., in wild animals, so this fills a valuable gap in knowledge.
I have a few comments regarding the manuscript, but they are only minor.
I believe, although it does get confusing, that all mentions of Salmonella need to be in italics as this is a Latin name for the family, but the subtypes are not italicised. Please do accept my ignorance if this is incorrect as its been a few years since I published in this field.
I am not sure that serotypization is a work? Perhaps serotyping maybe better?
Line 41- maybe change frequently to frequent?
Line 47- Gram needs capitalising
Line 48- role as a foodborne pathogen (reword)
Line 48- belongs to the … (Reword)
Line 140-141- perhaps reword this as its seems a bit unclear
Line 150- overall doesn’t need a capital letter
Line 153-166- perhaps this data may be better in a table? Its up to the authors who may have tried this. But it is just a thought as I read through
Figure 2- it may be my copy, but the colours seem to differ in their boldness- if that is the case could you make them all the same please?
Figure 3- perhaps make the text a bit larger here as its not very clear (in my copy anyway)
Line 169- space between 95% and credibility
Table 1- please use a full stop rather than a comma for the % values
Line 183-184- the text here appears to be a bit smaller than in other places?
Line 227- space needed between S and enterica
Line 236- is Afterwards the correct word? It suggests something after this study? Perhaps change?
Line 243- also similar for at last- maybe finally ?
Line 285- not sure why part of fourth is underlined?
